# Holographic Quantum Scars

Diego Liska [1*], Vladimir Gritsev [1,2], Ward Vleeshouwers [1,3], Jiří Minář [1,2,4]

**1** Institute for Theoretical Physics, Institute of Physics, University of Amsterdam,
Science Park 904, 1098 XH Amsterdam, the Netherlands
**2** QuSoft, Science Park 123, 1098 XG Amsterdam, the Netherlands
**3** QuiX Quantum B.V., Hengelosestraat 500, 7521 AN Enschede, The Netherlands
**4** CWI, Science Park 123, 1098 XG Amsterdam, the Netherlands

*d.liska@uva.nl,    j.minar@uva.nl

July 6, 2023

## Abstract

We discuss a construction of quantum many-body scars in the context of holography. We consider two-dimensional conformal field theories and use their dynamical symmetries, naturally realized through the Virasoro algebra, to construct scarred states. By studying their Loschmidt amplitude, we evaluate the states' periodic properties. A geometrical interpretation allows us to compute the expectation value of the stress tensor and entanglement entropy of these scarred states. We show that their holographic dual is related by a diffeomorphism to empty AdS, even for energies above the black hole threshold. We also demonstrate that expectation values in the scarred states are generally non-thermal and that their entanglement entropy grows with the energy as $\log(E)$ in contrast to $\sqrt{E}$ for the typical (bulk) states. Furthermore, we identify fixed points on the CFT plane associated with divergent or vanishing entanglement entropy in the limit where the scarred states have infinite energy.

# 1 Introduction

Following the formulation of the eigenstate thermalization hypothesis (ETH) [1–3], the problem of evolution towards thermal equilibrium has received considerable interest in the context of both quantum field theories and various condensed matter systems, most notably quantum spins on a lattice. In its strong version, the ETH stipulates that a generic energy eigenstate looks locally thermal, as quantified by the expectation values of quasi-local operators. In this form, the ETH is a statement about the matrix elements of such operators

$$\langle E_i | \mathcal{O} | E_j \rangle = f(\bar{E})\delta_{ij} + \mathrm{e}^{-S(\bar{E})/2} g(\bar{E}, \omega) R_{ij}, \tag{1}$$

where $|E_i\rangle, |E_j\rangle$ are the system's eigenstates, $\bar{E} = (E_i + E_j)/2$ and $\omega = E_j - E_i$ are the mean energy and energy difference respectively, $S(\bar{E})$ is the microcanonical entropy and $f$ and $g$ are two smooth functions related to the thermal one and two-point functions. Several aspects of thermalization have been recently addressed in the context of conformal field theories (CFT) and holography. This includes the thermal properties of the eigenstates [4–7], operator dynamics [8–10], matrix elements and OPE coefficients [11–13], and the related onset of quantum chaos and information scrambling [14–19].

Since the formulation of the ETH, a number of exceptions to this behavior have been identified. In addition to integrable models, which constitute a natural choice due to the extensive number of conserved quantities, ergodicity can be broken due to emergent conserved charges. Well known examples of ergodicity breaking phenomena described in the literature include *many-body localization* [20–28] and, more recently, the so-called *Hilbert space fragmentation* [29] and *quantum many-body scars* (QMBS) first formulated for lattice spin systems with kinetic constraints [30, 31].

Before dealing with further details, a natural question is if one can identify a similar kind of non-thermalizing behavior in the context of holography. In this paper, we address this question by focusing specifically on the phenomenon of quantum many-body scarring.

## 1.1 Quantum many-body scars

Observation of oscillatory behaviour and slow decay of spin correlations following a quench from certain initial states in a chain of Rydberg atoms [30] has triggered an intense effort in studies of what has become known as quantum many-body scars. The mechanism preventing fast thermalization has been attributed to a set of particular eigenstates, quantum many-body scars, which form an approximate energy ladder (similar to the spectrum of a harmonic oscillator), and are typically close to a product state [31, 32].

The terminology borrows from that of *single-particle quantum scars* first discussed by Heller in the Bunimovich stadium [33] (see also [34–37]); a chaotic system which, upon quantizing, exhibits eigenvalue statistics following (GOE) random matrix universality [38]. Upon numerically solving the Schrödinger equation in the Bunimovich stadium, Heller observed a concentration of certain high-lying energy eigenstates around unstable

(classical) periodic orbits. This is to be contrasted with 'generic' eigenstates of chaotic systems, which are locally indistinguishable from random superpositions of plane waves. In particular, these states constitute a violation of the ETH.

Turning back to the many-body setting, the observation of Ref. [30][1] has triggered a great interest in QMBS predominantly in condensed matter ranging from kinetically constrained to driven and disordered systems, see the recent reviews [40–43] and references therein. In contrast, and to the best of our knowledge, there seems to be relatively few systematic studies of scarring in the context of quantum field theories, some examples being a recent study adopting the single particle perspective [44], QMBS in Luttinger liquids [45,46], Schwinger model [47] or in the studies of quenches in a scalar $\phi^4$ theory [48], although the "scar" denomination has to be taken with some caution as we now discuss.

Since the field is still rapidly evolving, various notions of QMBS exist in the literature. Thus, we must specify what objects we call QMBS and what justifies such a denomination. An attempt at unifying the framework describing QMBS has been made recently in Refs. [49–51] by Moudgalya and Motrunich, who proposed the use of *commutant algebras* to characterize the QMBS exhaustively. The works [49–51], in turn, build on previous constructions of the scar eigenstates, namely that of Shiraishi and Mori [52] and Pakrouski et al. [53], where the latter defines scar states as singlets in the representations of certain Lie groups (this approach has been dubbed *group invariant* construction in [49]).

Loosely speaking, one distinguishes between *isolated* QMBS, like the ones found in the AKLT model [54–56], and scars generated by an underlying *dynamical symmetry* (also called a *spectrum generating algebra*). The latter admits one or more towers of QMBS and are the sole focus of our work.

## 1.2 Algebraic approach to weakly-nonergodic behavior

Here we briefly outline some possibilities for scars in quantum theories based on an underlying algebraic structure. From this family we explicitly exclude many exactly-solvable models (e.g. by Bethe ansatz) which are known to have many algebraic symmetries.

### 1.2.1 Dynamical symmetries and scars

The presence of dynamical symmetry operators was recognized as one of the generic features of scarring behavior in many-body quantum systems. We say that a system described by an operator $H$ has a dynamical symmetry (see, e.g., [57] for the early review of the dynamical symmetry in quantum mechanics) if there is an operator $Q$, or a family thereof, such that

$$[H, Q] = \omega Q, \tag{2}$$

where $\omega \in \mathbb{R}$. This dynamical symmetry implies three things. First, as soon as we know a single eigenstate $|\psi_0\rangle$ with energy $E_0$, one can construct a tower of states, with evenly spaced energy levels[2], with a repeated action of $Q$. Clearly, $Q$ acts as the raising operator in this tower of states. If $H$ is a Hermitian operator (typically a Hamiltonian), then $Q^\dagger$ plays the role of the lowering operator. This follows from the simple relation

$$[H, Q^\dagger] = -\omega Q^\dagger. \tag{3}$$

Second, the Heisenberg equation of motion implies that there are *persistent oscillations* with frequency $\omega$:

$$i\partial Q/\partial t = [H, Q] \quad \Rightarrow \quad Q(t) = Q(0)e^{i\omega t}. \tag{4}$$

---

[1]See also [39] for more recent experimental realization.
[2]This follows from the commutation relation $[H, Q^n] = n\omega Q^n$, since $HQ^n |\psi_0\rangle = (E_0 + n\omega) |\psi_0\rangle$.

Third, $H$ commutes with the commutator $[Q, Q^\dagger]$, which implies that

$$[Q, Q^\dagger] = P(H, \{N\}), \tag{5}$$

where $P(H, \{N\})$ is a polynomial of the Hamiltonian $H$ and a (possible) set of commuting charges $\{N\}$. There are many examples of this situation, including the Higgs algebra for a particle moving on a sphere in either central (Coulomb) or confining (oscillator) potentials [58, 59], the Schrödinger problem of a Coulomb particle on $S^3$ [60, 61] various models of nonlinear quantum optics [62] and spin systems, and systems related to the finite $\mathcal{W}$-algebras [63]. We shall see that the relation (5) is realized by states beyond the canonical scarred coherent states (cf. Sec. 2 for definition) as well as by possible scar candidates in higher dimensional CFTs as we discuss in Sec. 5.

The algebraic properties (2) and (5) provide a useful and powerful starting point for the analysis of the dynamics in a variety of settings and allow for straightforward generalizations. In particular this includes a class of quasi-exactly solvable models and Floquet scars which we now briefly discuss.

### 1.2.2  Quasi-exactly solvable models

A class of quasi-exactly solvable models was introduced in 80's by Turbiner and Ushveridze (for a review we refer to the paper by Turbiner [64] and to the book by Ushveridze [65]). The mathematical essence of these models is an existence of *invariant subspaces* of dimension $n$ which are preserved by the action of some (originally differential) operators $H : V_n \to V_n$. While in the generic case we would have to diagonalize an infinite-dimensional matrix to find the spectrum of the Hamiltonian of interest, in the case of quasi-exactly solvable models we are dealing with triangular matrices. The Hilbert space is stratified into the blocks such that the operator $H$ cannot move the states to the higher-dimensional space. In this case, part of the spectrum can be found exactly using algebraic methods. In most of the studies these models are associated with a hidden $sl(n)$ algebra structure.

To illustrate the above definitions, let's consider a particular $sl(2)$ example relevant to the present context. Introducing $L_{\pm,0}$ as a basis of $sl(2)$ one could attribute a *grading* of operators denoted by $deg$ such that $deg(L_{\pm,0}) = \pm 1, 0$. Then, $deg[L_+^{n_+} L_0^{n_0} L_-^{n_-}] = n_+ - n_-$. In this setup a *quasi-exactly solvable operator* belonging to the universal enveloping algebra of $sl(2)$ would have *no* terms of positive grading. This means it can not move states into a higher dimensional subspace in the flag $V_1 \subset V_2 \subset \ldots V_n \subset \ldots$. It follows that if the dynamics starts with a state which belongs to the $V_n$, the operator $H$, being regarded as a Hamiltonian, cannot escape the subspace of the flag contained in $V_n$. This implies that some part of the dynamics could have some finite-dimensional invariant subspaces, which could also be identified with scars. We note that many quasi-exactly solvable models are related to polynomially-deformed algebras discussed in Sec. (1.2.1). It is also known that quasi-exactly solvable models are intrinsically connected to conformal field theories, see [64, 66, 67].

### 1.2.3  Floquet scars

Suppose the operators $X$ and $Y$ are such that $[X, Y] = \omega Y$. From the Baker–Campbell–Hausdorff formula, it follows that

$$e^X e^Y = \exp\left(X + \frac{\omega Y}{1 - e^{-\omega}}\right). \tag{6}$$

This simple observation could be interpreted as follows. Let us define a discrete time two-step Floquet protocol, where the time evolution is given by the repeating sequence of products $U_F = e^{-iT_1 H_1} e^{-iT_2 H_2}$ with non-commuting $H_1$ and $H_2$. This implicitly defines what is called a Floquet Hamiltonian as $H_F = i \log(U_F)/T$ where $T = T_1 + T_2$. This Hamiltonian defines the stroboscopic (in terms of integer multiples of $T$) time evolution of the system. In most of the cases, the explicit evaluation of $H_F$ is a formidable task. However in certain (integrable) cases this can be done, see e.g. [68–70]. The situation defined by Eq. (6) belongs to this class of solvable Floquet systems. Moreover, this also defines what we could call *Floquet scars*. Indeed, if we identify one of the Hamiltonians with $Y$, that is $Y = -iT_2 H_2$, and the operator $X$ with $-iT_1 H_1$, we obtain the following Floquet Hamiltonian

$$H_F \equiv \frac{i}{T} \log(e^{-iT_1 H_1} e^{-iT_2 H_2}) = \frac{1}{T}\left(T_1 H_1 + \frac{\omega T_2 H_2}{1 - e^{-\omega}}\right). \tag{7}$$

This tells us that a suitable combination of $H_1$ and $H_2$ generates a scarred dynamics in the sense of a stroboscopic Floquet evolution $U_F^\dagger O U_F = e^{i\Omega T} O$ for some $O$ and $\Omega$. Further generalizations of these simple observations can be made with the use of Refs. [71–73] which treat more general solvable cases of the Baker-Campbell-Hausdorff formula than the relation (6).

In relation to the present paper we note that recently several studies discussed Floquet conformal field theories [74–77]. It seems natural to interpret the results of Refs. [74–77] in terms of the Floquet scars by identifying the corresponding Hamiltonians $H_1$ and $H_2$ (in this case given by a linear combination of the Virasoro generators $L_{\pm 1, 0}$) and exploiting the properties (6), (7). We leave this interesting opening for future investigations.

## 1.3 Formulation of the problem and relation to other works

On the CFT side, a well-known example of reviving states in 2d boundary CFTs was found by Cardy in [78] (see also [79, 80]). These are specific descendants of the ground state, which are created from a finite subspace of the plane. In this case, the periodicity arises from the finite size of the system, where particles moving at the speed of light bounce back and forth between the two (finitely separated) boundaries. Similar periodic behaviour has been also observed recently in the studies of information scrambling in 2d CFTs [81, 82] and in studies of quenches in marginally-deformed Tomonaga-Luttinger liquid building on the underlying free bosonic CFT [46].

On the gravity side, one can find various examples of periodic behavior ranging from studies of circular orbits in AdS black hole backgrounds [83] to discrete scale invariance [84, 85], cyclic RG flows [86], "solar systems" in AdS and its CFT dual [87] or time-periodic behaviour of a probe self-interacting scalar field, exhibiting a tower-of-states like structure, on the AdS background [88]. See also [89–92] for related topics.

In AdS$_3$, there is a mass threshold for the creation of black holes [93] given by $h \geq c/24$, where $h$ is the conformal dimension of the dual CFT operator and $c$ the central charge. The thermalization of the boundary CFT can be seen as equivalent to strong gravitational interactions in the boundary. Below the BTZ threshold, a CFT operator insertion creates conical defects, which gives rise to periodically reviving OTOCs [16]. This raises the question of whether there exist states above the BTZ threshold which do not thermalize but instead display periodic revivals.

We provide an affirmative answer to this question as follows: for CFTs endowed with the Virasoro algebra, one can identify the Hamiltonian in the usual way as $L_0 + \bar{L}_0$, the sum of holomorphic and antiholomorphic Virasoro generators. Realizing that the Virasoro algebra contains generators, which satisfy the dynamical symmetry condition

(2), we identify the scarred state as a generalized coherent state formed by acting with a displacement operator on a reference state – here a primary $|h\rangle$ of a scaling dimension $h$. This ensures that such a scarred state is formed solely by a superposition of scars in the sense of the definition of Eq. (2) and displays the characteristic periodicity generated by the Hamiltonian $L_0 + \bar{L}_0$ in the real-time evolution. Here, we would like to emphasize that in order to avoid ambiguity, we use the word *scar* to designate specific eigenstates while we use *scarred state* to designate a state, which can be written as a superposition of scars.

This paper is organized as follows. Focusing specifically on 2d CFTs, we introduce the relevant states and operators in Sec. 2. We then characterize the scarred state using its autocorrelation function (Loschmidt amplitude) in Sec. 2.1 and evaluate the expectation value of the stress-energy tensor in Sec. 3. This provides the input for the holographic description in $AdS_3$ where we focus specifically on the computation of entanglement entropy (EE) using the Ryu-Takayanagi prescription in Sec. 4.

Intriguingly, the diffeomorphism associated with the stress-energy tensor allows us to identify two classes of points on the CFT plane, which we dub *stable* and *unstable*, cf. Sec. 4 for the details. We find that the EE between a pair of points belonging to the different classes (stable or unstable) diverges in the limit where the scarred states have infinite energy, while the EE between a pair of points belonging to a different class is that of the vacuum.

We conclude in Sec. 5 and discuss possible extensions, including the construction of states beyond coherent states, scarred states in higher dimensions, and a link to minimal models and their deformations.

### Remarks

Before proceeding, we would like to point out that, recently, closely related studies have appeared in the literature. In Ref. [94], Dodelson and Zhiboedov address the problem of periodic orbits of a test mass in the AdS black hole background in general dimensions and link them to the scar states which the authors identify, in a perturbative limit, as the double-twist operators on the CFT side. This seems as a qualitatively different notion of scarring in contrast to the tower of scars contained in the generalized coherent state discussed here and with the interpretation of a diffeomorphism to/from empty $AdS_3$ on the gravity side.

Next, upon completion of our work, we became aware of another work by Caputa and Ge [95] with overlapping content. In order to emphasize the most notable differences, our work proceeds from the point of view of quantum many-body scars and their algebraic construction, emphasizing these scarred states' dynamics and non-thermal properties. We believe that the computation of the stress tensor presented here provides considerable technical simplifications (although the computation of the EE proceeds along similar lines following the standard Ryu-Takayanagi prescription). We also give an in-depth analysis of the interpretation of our scarred states as reparametrization, cf. Figs. 2 and 3. Finally, we focus on the EE of the scarred state with vacuum reference state; we provide further details about its structure, and identify a set of special points on the CFT plane with interesting properties as discussed in Sec. 4.

One more remark is about thermalization in 2d CFTs, where significant progress has been made regarding the thermal properties of primary and descendant states [96, 97]. Here, typical high-energy states have been identified and shown to exhibit thermal correlation functions [7, 98] [3]. Notably, at finite central charge, a typical microstate of weight

---

[3]It is known that in 2d CFTs the Hamiltonian $L_0$ is highly degenerate: all descendant states (of some primary $h$) at the same level $n$ have the same energy $h+n-c/24$. This leads to the ambiguity in defining the

$h$ is not a primary state but rather a descendant state of level $h/c$. Primary states only reproduce thermal correlation functions at large values of the central charge. To break away from thermality at large central charge, one must consider descendant states of level larger than $h/c$ or their linear combinations. Examples of such states are coherent states, which we construct explicitly and explain their lack of thermalization from the perspective of both the CFT and their gravitational dual.

## 2 Scars in 2d CFTs

Focusing on 2d CFTs, we first consider the standard Virasoro algebra[4]. In what follows, we only work with the holomorphic part of the theory. The whole discussion can then be straightforwardly extended to the antiholomorphic part as they commute.

$$[L_m, L_n] = (m - n)L_{m+n} + \frac{c}{12}m(m^2 - 1)\delta_{m+n,0}, \tag{8}$$

where $L_m$ are the Virasoro generators, and $c$ the central charge. We consider the customary choice for the Hamiltonian

$$H = L_0, \tag{9}$$

corresponding to the dilation operator. A general charge $Q$ satisfying (2) can then be written as a polynomial in $L_m$

$$Q = \sum_{\{m_i\}} \lambda_{\{m_i\}} \prod_{\{m_i\}} L_{m_i}, \quad \text{with} \quad \sum_i m_i = d, \tag{10}$$

where $\lambda_{\{m_i\}} \in \mathbb{C}$ and $d \in \mathbb{Z}$ is the degree of the charge. For example, the charge

$$Q = \lambda_{\{k\}} L_k + \lambda_{\{2k,-k\}} L_{2k} L_{-k}$$

satisfies (2) with $\omega = -d = -k$. We also note that for a Hermitian Hamiltonian, the conjugate of (2) implies

$$\left[ H, Q^\dagger \right] = -\omega Q^\dagger,$$

which, however, does not imply the antihermicity of the charge, i.e., $Q \neq -Q^\dagger$ in general. The antihermiticity of $Q$ can be imposed by modifying the expression (10) as

$$Q = \sum_{\{m_i\}} \lambda_{\{m_i\}} \prod_{\{m_i\}} L_{m_i} - \lambda^*_{\{m_i\}} \prod_{\{m_i\}} L_{-m_i}, \quad \text{with} \quad \sum_i m_i = d. \tag{11}$$

While $Q = -Q^\dagger$ is not required by (2), it is a convenient choice for the construction of the scarred states, as we now discuss. We define the scarred state as a generalized coherent state of the dynamical symmetry acting upon $|h\rangle$ as

$$|\lambda\rangle = D(\{\lambda\}) |h\rangle \equiv e^{-Q} |h\rangle, \tag{12}$$

---

ETH which can be avoided by using the integrable qKdV basis [99]. A generalized ETH then corresponds to thermalization with respect to a Hamiltonian deformed by a combination of the qKdV charges [96, 100]. In the present paper we however postulate that some form of thermalization is already there and we use $L_0$ as a Hamiltonian.

[4]It is also possible to consider Virasoro with an extended chiral algebra. In this setting, there are additional scarred states whose details depend on the enhanced symmetries of the theory. See, for example [46].

where $|h\rangle = \phi(0)|0\rangle$ is the state corresponding to a primary field $\phi(z)$ of scaling dimension $h$ and we have introduced the displacement operator $D$, which is unitary due to the antihermiticity of $Q$.

More specifically, the definition of the scarred state (12) corresponds to the definition of a generalized coherent state as introduced by Perelomov [101]. In principle, one could equivalently use a different definition of a coherent state, such as the Barut-Girardello state [102]; however, we shall adopt the former choice for the remainder of this article.

In order to simplify the treatment, we concentrate on the following special case of the antihermitian charge (11)

$$Q = \lambda L_{-k} - \lambda^* L_k \qquad (13)$$

and the corresponding displacement operator $D(\lambda)$. Here, we note that the action of the displacement operator $D(\lambda)$ on the primary can be evaluated with the help of the $su(1,1)$ subalgebra of the Virasoro algebra, which is obtained through the following identification

$$J_\pm = \frac{L_{\mp k}}{k} \qquad (14a)$$

$$J_0 = \frac{L_0}{k} + \frac{c_k}{2k^2}, \qquad (14b)$$

yielding the familiar algebra

$$[J_0, J_\pm] = \pm J_\pm, \quad [J_-, J_+] = 2\sigma J_0. \qquad (15)$$

In (14b) we used $c_k = \frac{c}{12}k(k^2-1)$ for the central extension of the Virasoro algebra (8), and in (15) $\sigma = 1$. Although in this work we are concerned solely with the $su(1,1)$ algebra, setting $\sigma = -1$ corresponds to the $su(2)$ algebra instead, cf. also the expressions (19), (20) below. Using (15) allows for further efficient manipulations. In particular, we introduce the following symmetric, normal, and anti-normal ordered forms:

$$U_S = \exp(a_+ J_+ + a_0 J_0 + a_- J_-) \qquad (16a)$$

$$U_N = \exp(A_+ J_+)\exp(\log[A_0]J_0)\exp(A_- J_-) \qquad (16b)$$

$$U_A = \exp(B_- J_-)\exp(\log[B_0]J_0)\exp(B_+ J_+), \qquad (16c)$$

These forms are equivalent given the following relations between variables (see e.g. [72,103] for a derivation)

$$A_\pm = \frac{\frac{a_\pm}{\mathcal{D}}\sinh\mathcal{D}}{\cosh\mathcal{D} - \frac{a_0}{2\mathcal{D}}\sinh\mathcal{D}}, \qquad A_0 = \left(\cosh\mathcal{D} - \frac{a_0}{2\mathcal{D}}\sinh\mathcal{D}\right)^{-2} \qquad (17)$$

$$B_\pm = \frac{\frac{a_\pm}{\mathcal{D}}\sinh\mathcal{D}}{\cosh\mathcal{D} + \frac{a_0}{2\mathcal{D}}\sinh\mathcal{D}}, \qquad B_0 = \left(\cosh\mathcal{D} + \frac{a_0}{2\mathcal{D}}\sinh\mathcal{D}\right)^{2} \qquad (18)$$

and

$$A_0 = \frac{B_0}{(1 - \sigma B_+ B_- B_0)^2}, \qquad A_\pm = \frac{B_\pm B_0}{1 - \sigma B_+ B_- B_0}, \qquad (19)$$

$$B_0 = \frac{(A_0 - \sigma A_+ A_-)^2}{A_0}, \qquad B_\pm = \frac{A_\pm}{A_0 - \sigma A_+ A_-}, \qquad (20)$$

where $\mathcal{D} = \frac{1}{2}(a_0^2 - 4\sigma a_+ a_-)^{1/2}$ and $\sigma = 1$ for $su(1,1)$.

## 2.1 Loschmidt amplitude

Equipped with the equations relating the forms in (16), we proceed with the computation of the Loschmidt amplitude (the autocorrelation function)

$$\mathcal{L}(t) = \langle \lambda(t) | \lambda(0) \rangle = \langle h | D(\lambda)^\dagger e^{itL_0} D(\lambda) | h \rangle . \tag{21}$$

This amplitude provides a measure of the scarring behavior of $|\lambda\rangle$ in that it depicts periodic revivals (by construction) in contrast to a simple decay (cf. also the discussion below for the relation to the orthogonality catastrophe). To evaluate $\mathcal{L}(t)$, we first rewrite the displacement operator $D(\lambda)$ in a normal ordered form. Then, when acting on the primary state $|h\rangle$, we find that

$$|\lambda\rangle = e^{-\lambda k J_+ + \lambda^* k J_-} |h\rangle = A_0^\mu e^{A_+ J_+} |h\rangle , \quad \text{where} \tag{22}$$

$$\mu = \frac{h}{k} + \frac{c}{24}\frac{k^2 - 1}{k}, \quad A_0 = \cosh^{-2}(k|\lambda|), \quad \text{and} \quad A_+ = -\frac{\lambda}{|\lambda|}\tanh(k|\lambda|). \tag{23}$$

The next step is to use the relations between anti-normal and normal ordered forms,

$$\begin{aligned}
\mathcal{L}(t) &= e^{-i\frac{c_k t}{2k}} A_0^{2\mu} \langle h | e^{A_+^* J_-} e^{itk J_0} e^{A_+ J_+} | h \rangle \\
&= e^{-i\frac{c_k t}{2k}} A_0^{2\mu} \left[ \frac{e^{itk}}{\left(1 - e^{itk}\tanh^2(k|\lambda|)\right)^2} \right]^\mu \\
&= e^{ith} \left( \frac{1}{\cosh^2(k|\lambda|)} \frac{1}{1 - e^{ikt}\tanh^2(k|\lambda|)} \right)^{2\mu}
\end{aligned} \tag{24}$$

where we have used the normalization condition $\langle h|h\rangle = 1$. From the Loschmidt amplitude, we can extract the energy of the scarred state using the relation,

$$\langle \lambda | L_0 | \lambda \rangle = \left. \frac{1}{i}\frac{\partial}{\partial t} \log \mathcal{L}(t) \right|_{t=0} = h + 2\left[ h + \frac{c}{24}(k^2 - 1) \right]\sinh^2(k|\lambda|). \tag{25}$$

Note that these states can have arbitrarily high energies as $\lambda$ goes to infinity. For large values of $\lambda \gg c$, the energy grows exponentially with $\lambda$, $E \sim e^{2k|\lambda|}$. This is relevant because we want to construct high-energy states that do not exhibit thermal properties.

In Fig. 1 we show the Loschmidt echo $|\mathcal{L}(t)|^2$ for several choices of $k, h, c$ and $\lambda$. By construction, the Loschmidt echo depicts perfect wavefunction revivals for times $t_m = 2\pi m/k$, $m \in \mathbb{Z}$. It is worth noting that for any time away from the perfect revival $t \neq t_m$ the Loschmidt echo is smaller than one and vanishes when any of the parameters $c, k, h$ or $|\lambda|$ is sent to infinity (while the others are kept fixed), in which case the Loschmidt echo resembles a Dirac comb.

From (24), it follows that for short times (or more specifically for times following the exact revivals), the Loschmidt amplitude decays as

$$|\mathcal{L}(t)| \approx e^{-\frac{\mu}{4}(kt)^2\sinh(2\lambda k)} = e^{-\left(\frac{t}{t_c}\right)^2}, \tag{26}$$

where one identifies the characteristic decay time as

$$t_c = \frac{2}{k\sqrt{\mu\sinh(2\lambda k)}}. \tag{27}$$

We thus see, that for large values of $h, c$, the Loschmidt amplitude decays exponentially with exponent proportional to $h, c$, and the time-evolved state becomes quasi-orthogonal

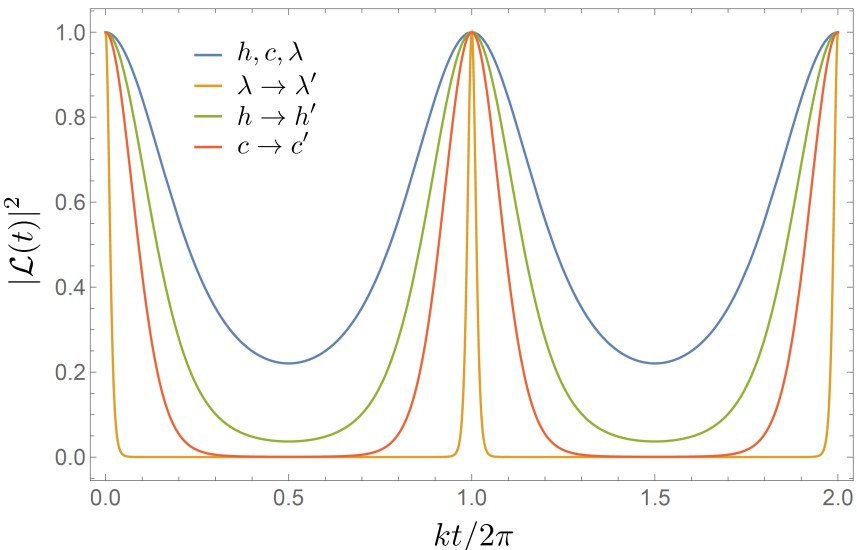

Figure 1: Loschmidt echo $|\mathcal{L}(t)|^2$, cf. Eq. (24), for $k = 3$. The blue line corresponds to the vacuum state with the set of parameters $(h, c, \lambda) = (0, 20, 0.1)$. The orange, green and red lines show the effect of varying $\lambda, h$ and $c$ as $\lambda' = 5\lambda$, $h' = 10c/24$, i.e. a value above the BTZ threshold, and $c' = 5c$ respectively. The dependence for the threshold value $h = c/24$ resembles that of $h = 0$ and is not shown.

to the initial state (a feature known as orthogonality catastrophe [104,105]) after the time scale $t_c$. This should be compared to the situation encountered in many-body systems, such as a quenched Luttinger liquid [106] (see also [107]), where the exponent scales proportionally to the system size. In this sense the result (26) corroborates with this observation in that $h$ and $c$ quantify the number of degrees of freedom in the CFT.

Further properties of the Loschmidt amplitude are as follows. For large amplitude $\lambda$ of the scarred state, it decays double exponentially as $\exp(-\exp(\lambda k))$, up to multiplicative factors. For large values of $k$, it decays as $\sim \exp(-k^3 \exp(\lambda k))$. Finally, parametrizing the scaling dimension as $h = \epsilon \frac{c}{24}$, we can rewrite the exponent of (24) as

$$\mu = \frac{h}{k} + \frac{c}{24} \frac{k^2 - 1}{k} = \frac{c}{24} \left( k + \frac{\epsilon - 1}{k} \right), \tag{28}$$

where $\epsilon = 1$ corresponds to the BTZ threshold. For $k > 1$ assumed here, the sub (super) threshold value of $\epsilon < 1$ ($\epsilon > 1$) thus corresponds to the sign change of the $1/k$ correction to the exponent of the Loschmidt amplitude.

**Symmetries**

We can consider an alternative definition of the (non-normalized) scarred state based on a charge that is not antihermitian $\tilde{Q} = \tilde{\lambda} L_{-k}$: $|\tilde{\lambda}\rangle = \mathrm{e}^{-\tilde{Q}} |h\rangle$. In this case, an analogous derivation as that of (24) leads to (see Appendix A for the details)

$$\mathcal{L}(t) = \left[ \frac{1 - |\tilde{\lambda}k|^2}{1 - |\tilde{\lambda}k|^2 \mathrm{e}^{ikt}} \right]^{2\mu} \mathrm{e}^{iht}. \tag{29}$$

This expression is invariant, up to an overall phase, with respect to the replacement $|\tilde{\alpha}| \to 1/|\tilde{\alpha}|$, where $\tilde{\alpha} = \tilde{\lambda}k$. This can be interpreted as a weak-strong drive[5] symmetry of

---

[5]Here we borrow the language from quantum optics where $D(\alpha) |0\rangle = |\alpha\rangle \propto \mathrm{e}^{\alpha a^\dagger} |0\rangle$ corresponds to a coherent state of amplitude $\alpha$ with $a^\dagger$ a bosonic creation operator, used to describe driving a system with

the Loschmidt amplitude. It is interesting to link the symmetry in (29) to the expression (24) based on the scarred state using the antihermitian charge. To this end we note that

$$
\begin{aligned}
e^{\alpha J_+ - \alpha^* J_-} |h\rangle &= e^{\tilde{\alpha} J_+} e^{\log \cosh^{-2} |\alpha| J_0} e^{-\tilde{\alpha}^* J_-} |h\rangle \\
&= e^{\tilde{\alpha} J_+} |h\rangle \cosh^{-2\mu} |\alpha|
\end{aligned}
\tag{30}
$$

with $\alpha = \lambda k$ and

$$
\tilde{\alpha} = \frac{\alpha}{|\alpha|} \tanh |\alpha|.
\tag{31}
$$

Consequently, the symmetry of the Loschmidt amplitude (29) under the replacement $|\tilde{\alpha}| \to 1/|\tilde{\alpha}|$ translates into the symmetry of (24) under the replacement $\tanh |\alpha| \to 1/\tanh |\alpha|$ (and the associated change $\cosh |\alpha| \to i \sinh |\alpha|$). We also note that while the amplitude of the scarred state in (29) can take any non-negative values, $|\tilde{\alpha}| = |\tilde{\lambda} k| \geq 0$, the identification between $\alpha$ and $\tilde{\alpha}$ restricts the values of $\tilde{\alpha}$ to $|\tilde{\alpha}| \leq 1$. This is a consequence of requiring the absence of $J_0$ in the definition of the unitary displacement operator, i.e. in the exponent of the left-hand-side of (30).

## 3    Scars as reparameterizations

The scarred states are built from unitary representations of the Virasoro algebra. This group is the generator of coordinate transformation on the complex plane, meaning that, we can interpret these scarred states as coordinate reparametrizations. The aim of this section is to make this connection more precise. To start, take a primary operator $\mathcal{O}(z)$ on the complex $z$-plane with conformal weight $h_{\mathcal{O}}$, and consider the action of the displacement operator $D(\lambda)$ when $\lambda \ll 1$,

$$
D(\lambda)^\dagger \mathcal{O}(z) D(\lambda) = \mathcal{O}(z) + [Q, \mathcal{O}(z)] + \mathcal{O}\left(\lambda^2\right).
\tag{32}
$$

Using the commutation relations

$$
[L_k, \mathcal{O}(z)] = z^{k+1} \partial_z \mathcal{O}(z) + h_{\mathcal{O}}(k+1) z^k \mathcal{O}(z)
\tag{33}
$$

we find that, infinitesimally, this operator transforms like

$$
[Q, \mathcal{O}(z)] = h_{\mathcal{O}} \left[ \lambda(1-k) z^{-k} - \lambda^*(1+k) z^k \right] \mathcal{O}(z) + \left( \lambda z^{1-k} - \lambda^* z^{k+1} \right) \partial_z \mathcal{O}(z).
\tag{34}
$$

This expression corresponds to the infinitesimal transformation of a primary operator under the change of coordinates $z \to w = z + \lambda z^{1-k} - \lambda^* z^{1+k} + \mathcal{O}\left(\lambda^2\right)$ from the complex plane to itself.

To find the full map from $z$ to $w$, we can consider instead the exponentiated version of (34) [108]

$$
D(\lambda)^\dagger \mathcal{O}(z) D(\lambda) = \left( \frac{\partial f(\lambda, \lambda^*, z)}{\partial z} \right)^{h_{\mathcal{O}}} \mathcal{O}\left(f(\lambda, z)\right),
\tag{35}
$$

where the function $w(z) = f(\lambda, \lambda^*, z)$ is the map that we want to determine. To simplify the discussion, we consider the case where $\lambda$ is imaginary so that $\lambda = -\lambda^*$ is the only parameter. Taking the derivative with respect to $\lambda$ on both sides of (35), and using the commutation relation (33), we arrive at the differential equation

$$
\partial_\lambda f(\lambda, z) = (z^{1+k} + z^{1-k}) \partial_z f(\lambda, z).
\tag{36}
$$

---

a laser of intensity $\propto |\alpha|^2$.

This equation, when supplemented with the initial condition $f(0, z) = z$, has as a solution

$$f(\lambda, z) = \tan\left(k\lambda + \arctan(z^k)\right)^{\frac{1}{k}} = \left(\frac{\eta + z^k}{1 - \eta z^k}\right)^{\frac{1}{k}}, \tag{37}$$

where we have defined the new variable $\eta = \tan(k\lambda)$.

An important remark is that $f$ is not single-valued. There are $k$ different roots for a given value of $z$ and $\eta$. Nonetheless, there is a natural choice that singles out one of these roots. Using the initial condition $f(0, z) = z$, the prescription is to pick the root continuously connected to $z$ when $\eta = 0$. This point will be important when we discuss the entanglement properties of the states in Sec. 4.

This analysis shows that correlation functions in the scarred state $|\lambda\rangle$ can be computed in the reference state $|h\rangle$ using a change of coordinates, $z \to w = f(\lambda, z)$. For example, the $n$-point function of $n$ primaries $\mathcal{O}_i$ with conformal dimensions $h_{\mathcal{O}_i}$ on the $z$-plane is given by

$$\langle\lambda| \mathcal{O}_1(z_1)\mathcal{O}_2(z_2)\cdots\mathcal{O}_n(z_n) |\lambda\rangle = \langle h| D(\lambda)^\dagger\mathcal{O}_1(z_1)D(\lambda)D(\lambda)^\dagger\mathcal{O}_2(z_2)\cdots D(\lambda) |h\rangle$$

$$= \prod_i \left(\frac{\partial f(\lambda, z)}{\partial z}\right)^{h_{\mathcal{O}_i}}_{z=z_i} \langle h| \mathcal{O}_1\big(f(\lambda, z_1)\big)\cdots\mathcal{O}_n\big(f(\lambda, z_n)\big) |h\rangle. \tag{38}$$

In the rest of this paper, we will use the variable $z$ to evaluate correlation functions in the scarred state $|\lambda\rangle$, and the variable $w$ for correlation functions in the reference state $|h\rangle$ (see Fig. 2). We will also make a distinction between primary operators on the $z$-plane, denoted by $\mathcal{O}(z)$, and primary operators on the $w$-plane, denoted with a prime $\mathcal{O}'(w)$. The primed operators are defined as

$$\mathcal{O}'(w) = D(\lambda)\mathcal{O}(w)D(\lambda)^\dagger, \tag{39}$$

so that they reproduce the familiar transformation rule of a primary field,

$$\mathcal{O}'(w(z)) = \left(\frac{\partial w}{\partial z}\right)^{-h_{\mathcal{O}}} D(\lambda)\left(D(\lambda)^\dagger\mathcal{O}(z)D(\lambda)\right)D(\lambda)^\dagger = \left(\frac{\partial w}{\partial z}\right)^{-h_{\mathcal{O}}}\mathcal{O}(z). \tag{40}$$

Before moving to the computation of the stress tensor, we would like to comment on a straightforward generalization of this analysis. Consider the operator

$$D(\lambda) = \exp\left(-\lambda\sum_k a_k L_k\right) \tag{41}$$

with $\lambda \in i\mathbb{R}$ and $a_k \in \mathbb{R}$. When this operator acts on a primary state, it leads to the transformation rule

$$\sum_k a_k\big(h_{\mathcal{O}}(k+1)z^k + z^{1+k}\partial_z\big)\left[(\partial_z g(\lambda, z))^{h_{\mathcal{O}}}\,\mathcal{O}\big(g(\lambda, z)\big)\right]$$

$$= \partial_\lambda\left[(\partial_z g(\lambda, z))^{h_{\mathcal{O}}}\,\mathcal{O}\big(g(\lambda, z)\big)\right], \quad (42)$$

that results in the following differential equation,

$$\frac{\partial_z g(\lambda, z)}{\partial_z G(z)} = \partial_\lambda g(\lambda, z), \quad \text{with} \quad \frac{1}{\partial_z G(z)} = \sum_k a_k z^{1+k}. \tag{43}$$

Solutions to this equation are of the form

$$g(\lambda, z) = G^{-1}\big(\lambda + G(z)\big), \tag{44}$$

where $G(z)$ is defined up to a constant term. One can check that the special case of $a_k = a_{-k} = 1$ corresponds to the transformation (37).

## 3.1 The stress tensor

Knowing the map associated with the scarred state $|\lambda\rangle$, we can compute the expectation value of the stress tensor using the familiar transformation rule

$$T'(w(z)) = \left(\frac{dw}{dz}\right)^{-2}\left[T(z) - \frac{c}{12}\{w(z),z\}\right]. \tag{45}$$

Here, the Schwarzian derivative is given by the expression,

$$\{w,z\} = \frac{w'''(z)}{w'(z)} - \frac{3}{2}\left(\frac{w''(z)}{w'(z)}\right)^2. \tag{46}$$

More explicitly, the stress tensor of the scarred state is given by[6]

$$\langle\lambda|\,T(z)\,|\lambda\rangle = \left(\frac{dw}{dz}\right)^2\frac{h}{w(z)^2} + \frac{c}{12}\{w(z),z\}$$
$$= \frac{24\left(\eta^2+1\right)^2 h z^{2k} - c\eta\left(k^2-1\right)\left(2\eta z^k + z^{2k}-1\right)\left(z^k\left(\eta z^k-2\right)-\eta\right)}{24 z^2\left(\eta+z^k\right)^2\left(\eta z^k-1\right)^2}. \tag{47}$$

This computation makes it clear that the displacement operator $D(\lambda)$ simply moves the stress tensor of the reference state $|h\rangle$ along the same *Virasoro coadjoint orbit*: two elements or states are said to lie within the same orbit if there is a holomorphic transformation, in this case $z \to w$, relating them. One can think of each orbit as containing the set of a primary operator together with its Virasoro descendants. These orbits are classified using the expectation value of the stress tensor and its transformation properties. For a review of this topic, we recommend [109, 110] and references therein. Interestingly, these orbits have energies that are unbounded from above. For us, this is reflected in the fact that we can prepare scarred states with arbitrarily high energies that do not exhibit thermal properties, as we now discuss.

## 3.2 Thermalization

The eigenstate thermalization hypothesis successfully describes how a single eigenstate of a, usually chaotic, many-body quantum system can be considered as being in thermal equilibrium [1–3]. In this setting, the ETH is a statement about the matrix elements of a thermalizing observable. We say that an operator $\mathcal{O}$ thermalizes with respect to the eigenstates $|E_i\rangle$ if the expectation values $\langle E_i|\,\mathcal{O}\,|E_i\rangle$ are given in terms of a function, that depends smoothly on the energy $E_i$, and corresponds to the thermal expectation value of $\mathcal{O}$, with possibly very small erratic fluctuations. That is

$$\langle E_i|\,\mathcal{O}\,|E_i\rangle = \text{Tr}(\rho_{\text{th}}\mathcal{O}) + \mathcal{O}\left(\text{e}^{-S(E_i)/2}\right), \tag{48}$$

where, $\rho_{\text{th}}$ is the thermal density matrix $\text{e}^{-\beta H}/\text{Tr}(\text{e}^{-\beta H})$, whose inverse temperature $\beta$ is the one associated to the energy of the eigenstate $|E_i\rangle$. Here, the $S(E_i)$ corresponds the microcanonical entropy. The full formulation of ETH also includes the behavior of off-diagonal matrix elements, see Eq. (1), but in this section, we will focus on the diagonal elements only.

The most unclear part of the ETH is the class of operators and eigenstates to which it applies. There are obvious cases where it fails, the typical example being that of a

---

[6]We note that the expression Eq. (3.3) found in [95] matches the result (47) upon the substitution $-\eta \to z_k, \eta \to \bar{z}_k$ in the latter.

projector operator $|E_i\rangle \langle E_i|$. In the context of 2d CFTs[7], the ETH is believed to work well on light primary operators when the spectrum is restricted to only primary eigenstates [6, 11, 16, 113]. Moreover, the ETH is expected to hold only for heavy primary eigenstates $h \gg c$.

Unfortunately, in the context of CFTs, a simple computation reveals that at large enough energies, most states are not primaries [7]. In fact, at a given fixed energy $h/c \gg 1$, the fraction of states which are primaries is exponentially suppressed

$$\frac{\#(\text{primary states})}{\#(\text{all eigenstates})} \sim e^{-\pi \sqrt{\frac{h}{6c}}}. \tag{49}$$

At these energies, the relevant question is not whether ETH holds for a primary state but if it holds for a typical eigenstate. This question was explored in detail in [7], for the regime in which $h \to \infty$ with $h/c$ fixed, the authors observed ETH-like behavior for the matrix elements of simple operators with respect to typical states[8]. They found that diagonal matrix elements lie along a smoothly varying curve, and off-diagonal elements are exponentially suppressed relative to the diagonal ones.

In this paper, we have created a one-parameter family of states parametrized by $\eta$. These states have arbitrarily high energies, and we would like to know if an operator that obeys ETH in a given energy window will also appear thermal when evaluated in the scarred state. The answer is, as expected, that this is not the case. In fact,

$$\langle \lambda | \mathcal{O}(1) | \lambda \rangle = \left( \frac{1 - |\eta|^2}{1 + |\eta|^2} \right)^{h_{\mathcal{O}}} C_{h\mathcal{O}h} = \frac{\langle h | \mathcal{O} | h \rangle}{\cosh 2k|\lambda|}, \tag{50}$$

the expectation value of the scarred state is continuously connected to that of the reference state $|h\rangle$, and it differs from the thermal expectation value $\text{Tr}(\rho_{\text{th}}\mathcal{O})$ evaluated at the same energy $\langle \lambda | L_0 | \lambda \rangle$. More generally, the scarred states do not exhibit many of the thermal properties that have been established for, e.g., primary states. For example, in [16], the authors showed that for sparse, large-$c$ 2d CFTs,

$$\langle h, \bar{h} | \mathcal{Q}_1(z_1)\mathcal{Q}_2(z_2) \cdots \mathcal{Q}_n(z_n) | h, \bar{h} \rangle = \text{Tr}(\rho_{\text{th}}\mathcal{Q}_1(z_1)\mathcal{Q}_2(z_2) \cdots \mathcal{Q}_n(z_n)) + \mathcal{O}\left(c^{-1}\right), \tag{51}$$

where $\mathcal{Q}_1(z_1)$ are scalar primary operators with $h \gg h_{\mathcal{Q}_i}$. This result only holds if both $h$ and $\bar{h}$ are above the BTZ threshold of $c/24$. We can prepare a scarred state with the same energy $h + \bar{h}$, whose reference state is below this threshold; such a state will not exhibit this thermal property and will end up evaluating to a non-thermal $(n + 2)$-point function.

## 4  Entanglement

For large values of the central charge, we can use the Ryu and Takayanagi (RT) prescription to study the entanglement entropy (EE) of the scarred states. The RT prescription relates the length of a geodesic in an asymptotically $\text{AdS}_3$ space with the EE of its boundary subregion [114, 115]. In the Fefferman-Graham gauge, the relevant $\text{AdS}_3$ metric has the form

$$ds^2 = \frac{dr^2}{r^2} + r^2 dud\bar{u} - L(u)du^2 - \bar{L}(\bar{u})d\bar{u}^2 + \frac{1}{r^2}L(u)\bar{L}(\bar{u})dud\bar{u}. \tag{52}$$

---

[7]The ETH is expected to hold only for CFTs that display level repulsion in their spectrum of primary operators. This is expected to be equivalent to having a central charge $c > 1$, a sparse spectrum of light primary fields, and no extended Virasoro algebra. For more details about these conditions, we recommend the Refs. [11, 111, 112]

[8]In this context, a typical state at energy $h$ is a level $h/c$ descendant of a dimension $(c-1)h/c$ primary.

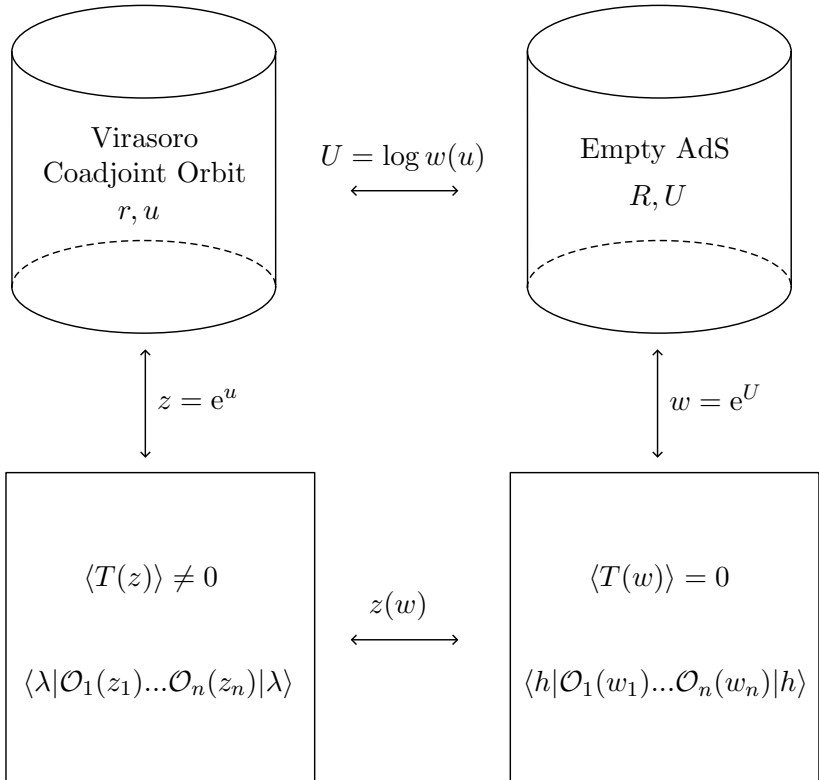

Figure 2: The scarred state $|\lambda\rangle = \mathrm{e}^{-Q}\,|h\rangle$ parametrized on the complex $z$-plane gets transformed under the map (37) to the reference state $|h\rangle$ on the complex $w$-plane. Correlation functions in the scarred state can be evaluated in the reference state using this map. Further transformations relate correlation functions on the plane to correlation functions on the cylinder.

In this section, we work in Euclidean time, so this metric has a positive signature. The variables $u$ and $\bar{u}$ are related to the complex coordinates $z$ and $\bar{z}$ of the CFT via the map from the cylinder to the plane, see Fig. 2,

$$z = e^u, \qquad \bar{z} = e^{\bar{u}}. \tag{53}$$

The relationship between this geometry and a CFT state is given by the identification [116]

$$L(u) = \frac{6}{c} \langle \lambda | T_{\text{cyl.}}(u) | \lambda \rangle, \qquad \bar{L}(\bar{u}) = \frac{6}{c} \langle \lambda | \bar{T}_{\text{cyl.}}(\bar{u}) | \lambda \rangle \tag{54}$$

where $T_{\text{cyl.}}(u)$ is the stress tensor of the CFT on the cylinder. This is the most general asymptotically AdS$_3$ solution to the Einstein equations whose boundary corresponds to the flat CFT cylinder. Different functions of $L$ and $\bar{L}$ give rise to a plethora of geometries: BTZ black holes, spinning particles, geometries with conical defects, etc[9]. In this paper, we focus on a very particular type of geometry, in which $L$ and $\bar{L}$ can be written as the Schwarzian derivative of a function $f(u)$:

$$L(u) = \frac{1}{2}\{f(u), u\}, \quad \text{and} \quad \bar{L}(\bar{u}) = \frac{1}{2}\{f(\bar{u}), \bar{u}\}. \tag{55}$$

This geometry will be relevant for us in the next section, where we study the EE of the scarred states.

We can simplify (52) by doing a change of coordinates. We have seen that there is a map that takes the nonzero stress tensor of the scarred state to a set of coordinates where the stress tensor vanishes. This map happens at the boundary of the AdS$_3$ geometry. Fortunately, there is a unique extension of this map into the radial direction $r$ that also preserves the Fefferman-Graham gauge. As an expansion in the $r$ variable, this map reads

$$\begin{aligned} u \to U &= h(u) - \frac{1}{2r^2} \frac{\partial_{\bar{u}}^2 h(\bar{u}) \partial_u h(u)}{\partial_{\bar{u}} h(\bar{u})} + \mathcal{O}\left(r^{-4}\right), \\ r \to R &= \frac{r}{\sqrt{\partial_u h(u) \partial_{\bar{u}} h(\bar{u})}} + \mathcal{O}\left(r^{-1}\right), \\ \bar{u} \to \bar{U} &= h(\bar{u}) - \frac{1}{2r^2} \frac{\partial_u^2 h(u) \partial_{\bar{u}} h(\bar{u})}{\partial_u h(u)} + \mathcal{O}\left(r^{-4}\right). \end{aligned} \tag{56}$$

For the computation of EE, we only need to know the first few terms in this expansion. Acting upon the metric (52) by the above transformation, one can check that the function $L(u)$ mimics the transformation properties of the CFT stress tensor

$$\tilde{L}\big(h(u)\big) = (\partial_u h(u))^{-2} \left(L(u) - \frac{1}{2}\{h(u), u\}\right). \tag{57}$$

To simplify the metric (52), we simply have to take $h(u) = \log f(u)$. The resulting change of coordinates yields a constant stress tensor and the metric

$$ds^2 = \frac{dR^2}{R^2} + R^2 dU d\bar{U} + \frac{dU^2}{4} + \frac{d\bar{U}^2}{4} + \frac{dU d\bar{U}}{16R^2}. \tag{58}$$

The RT prescription instructs us to compute the length of a geodesic with a given UV cut-off $r_0 \gg 1$, or equivalently,

$$R_0 = r_0 \sqrt{\frac{f(u)f(\bar{u})}{\partial_u f(u) \partial_{\bar{u}} f(\bar{u})}}. \tag{59}$$

---

[9]See [117] for a reference on the different geometries that arise from this metric.

We can quickly find solutions to the geodesic equation using the embedding formalism into Minkowski space[10]

$$
\begin{aligned}
x^0 &= \frac{4R^2 + 1}{4R} \cosh\left(\frac{U + \bar{U}}{2}\right), & x^1 &= \frac{4R^2 + 1}{4R} \sinh\left(\frac{U + \bar{U}}{2}\right), \\
x^2 &= \frac{1 - 4R^2}{4R} \cos\left(\frac{U - \bar{U}}{2i}\right), & x^3 &= \frac{1 - 4R^2}{4R} \sin\left(\frac{U - \bar{U}}{2i}\right).
\end{aligned}
\tag{60}
$$

In these coordinates, the hyperbolic distance $\ell(x_1, x_2)$ between two points $x_1$ and $x_2$ is given by the expression,

$$
\cosh \ell(x_1, x_2) = x_1^0 x_2^0 - x_1^1 x_2^1 - x_1^2 x_2^2 - x_1^3 x_2^3.
\tag{61}
$$

The geodesic distance between two points $u_1$ and $u_2$ at the boundary of the CFT, to first order in $r_0$ is then given by the relatively simple formula

$$
S(u_1, u_2) = \lim_{r_0 \to \infty} \frac{c}{6} \, \ell(u_1, u_2) = \frac{c}{6} \log r_0^2 \left| \frac{\left(f(u_1) - f(u_2)\right)^2}{\partial_u f(u_1) \partial_u f(u_2)} \right|,
\tag{62}
$$

where by $\partial_u f(u_1)$ we mean $\partial_u f(u)|_{u=u_1}$. Here, we used the RT formula that relates the length of the geodesic with the EE of the boundary subregion[11].

## 4.1 Scarred states

We now study the EE of the scarred state generated by the displacement of the vacuum

$$
|\lambda\rangle = D(\lambda) |0\rangle, \quad \text{with} \quad \langle T_{\text{cyl.}}(u) \rangle = \{f(\eta, e^u), u\},
\tag{63}
$$

and $f(\eta, u)$ given in (37). We rewrite $f$ here for convenience

$$
f(\eta, e^u) = \left(\frac{i|\eta| + e^{uk}}{1 - i|\eta|e^{uk}}\right)^{\frac{1}{k}}, \qquad |\eta| = \tanh(k|\lambda|).
\tag{64}
$$

Recall that this function is valid only when $\lambda$ or $\eta$ are imaginary numbers.

In the previous section, we remarked that this function is not single-valued due to its fractional power. There are $k$ different choices for the root $1/k$ at any given value of $\eta$ and $u$. A natural choice, however, is to pick the root that is continuously connected to the identity map as a function of $\eta$, that is, the root such that the function $f$ varies continuously from $f(0, u) = u$ to $f(\eta, u)$ as the norm of $\eta$ increases. This choice is consistent with the interpretation of the displacement operator as a unitary transformation satisfying $D(0) = \mathbb{1}$.

Having a clear definition of the function $f$, we now study its properties. First, note that the function leaves the unit circle invariant, $|f(\eta, e^{i\phi})| = 1$. Also, this map has $2k$ fixed points $\phi^\star$, defined by the property $f(\eta, e^{i\phi^\star}) = e^{i\phi^\star}$ and given by the angles

$$
\phi^\star = \frac{\pi}{2k}(4n \pm 1).
\tag{65}
$$

We can quickly see this from the differential equation that defines $f$,

$$
\partial_\lambda f(\lambda, e^{i\phi^\star}) = \left(e^{i\phi^\star(1+k)} + e^{i\phi^\star(1-k)}\right) \partial_z f(\lambda, z) = 0
\tag{66}
$$

---

[10]For a reference on these coordinate systems and the embedding formalism see [118].

[11]Formula (62) can also be derived from the CFT point of view. Using the replica trick, (62) follows from the two-point function of twist operators inserted at the ends of the interval. For more details on this computation, we refer the reader to [95].

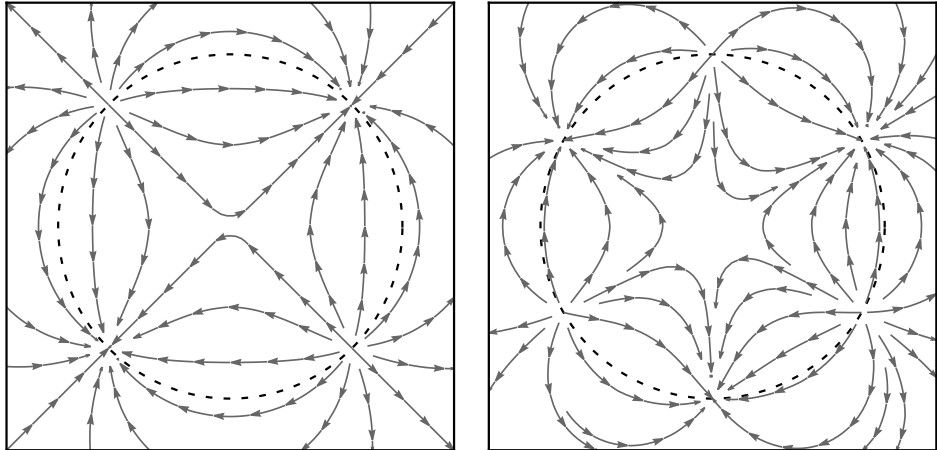

Figure 3: Two vector flows induced by the transformation $f(\eta, z)$ for $k = 2$ and 3. The unit circle is left invariant under this map and contains $2k$ fixed points. These fixed points are either stable or unstable with respect to the transformation induced by continuously increasing the value of $|\eta|$. The stable fixed points correspond to the $k$ roots of $i$, i.e., the points satisfying $z^k = i$. The unstable fixed points correspond to the $k$ roots of $-i$. The entanglement entropy of the regions bounded by these points, Eq. (71), simplifies drastically.

Half of these points are stable fixed points, and half are unstable fixed points with respect to the transformation induced by continuously increasing the value of $|\eta|$. These fixed points correspond to the roots of $\pm i$, i.e., $f(\eta, e^{u\phi^\star})^k = \pm i$. An easy method to distinguish between stable and unstable points is to do an expansion of $f(\eta, e^{i\phi})^k$ around $\phi = \phi^\star + \delta\phi$. We find that

$$f\left(\eta, \exp\left[\frac{\pi i}{2k}(4n+1) + i\delta\phi\right]\right)^k = +i - k\frac{1 - |\eta|}{1 + |\eta|}\delta\phi + \mathcal{O}(\delta\phi^2) \tag{67}$$

$$f\left(\eta, \exp\left[\frac{\pi i}{2k}(4n-1) + i\delta\phi\right]\right)^k = -i + k\frac{1 + |\eta|}{1 - |\eta|}\delta\phi + \mathcal{O}(\delta\phi^2). \tag{68}$$

This simple computation reveals that the roots of $i$ are stable fixed points for increasing $|\eta|$, and that the roots of $-i$ are unstable fixed points. In Fig. 3, we show an example of the flow generated by $f$ for $k = 2$ (left panel) and $k = 3$ (right panel).

The final component that we need to compute in order to evaluate EE is the magnitude of the derivative of the function $f$ with respect to $u$. Since the magnitude is a positive real number, this function is single-valued. When evaluated in the unit circle, this derivative has a relatively simple form

$$|\partial_u f(\eta, e^{i\phi})| = \frac{1 - |\eta|^2}{1 + |\eta|^2 + 2|\eta|\sin(k\phi)}. \tag{69}$$

For regions on the unit circle that are bounded by the fixed points of the map $f$, the expression for the EE simplifies drastically, and it is given by the formula

$$\Delta S(\eta) = S(\eta) - S(0)$$
$$= \frac{c}{6}\log\left(\frac{1 + |\eta|^2 + 2|\eta|\sin k\phi_1^\star}{1 - |\eta|^2}\right) + \frac{c}{6}\log\left(\frac{1 + |\eta|^2 + 2|\eta|\sin k\phi_2^\star}{1 - |\eta|^2}\right), \tag{70}$$

where, to avoid UV divergences, we have subtracted the vacuum EE $S(0)$. In terms of the original variable $\lambda = \arctan(\eta)/k$, we find the following possible outcomes

$$\Delta S(\lambda) = \frac{2c}{3} \begin{cases} k|\lambda|, & \text{stable} - \text{stable} \\ 0, & \text{stable} - \text{unstable} \\ -k|\lambda|, & \text{unstable} - \text{unstable} \end{cases} . \tag{71}$$

The entropy behaves differently depending on whether the subregion ends at stable or unstable fixed points[12].

Surprisingly, the entropy between unstable points decreases indefinitely, that is, with respect to the vacuum EE. Meanwhile, the EE between stable points behaves in the opposite way. If the region starts and ends at fixed points of different nature, we recover the EE of the vacuum state. We would like to remark that the choice of points on the unit circle changes the properties of the entanglement entropy. This is because the map $f$ breaks the $U(1)$ symmetry of the circle into a discrete $\mathbb{Z}_k$ symmetry.

Lastly, we would like to compare the EE of a region bounded by two stable fixed points with that of a primary state of the same energy. As a function of the energy, using (25), we have that the entropy of the scarred states behaves like [13]

$$\Delta S(E) \approx \frac{c}{3k} \log \left[ \frac{12E}{c(k^2 - 1)} \right]. \tag{72}$$

On the other hand, the EE of a primary state, with zero spin, $h = \bar{h}$, and conformal dimension $h + \bar{h} = E + c/12$, is given by the expression [121]

$$S_h(l) = \frac{c}{3} \log \left[ \frac{\beta_h}{\pi \epsilon_{\text{UV}}} \sinh \frac{l\pi}{\beta_h} \right], \quad \beta_h = \frac{2\pi}{\sqrt{\frac{24h}{c} - 1}}. \tag{73}$$

Here, $l$ is the length of a segment (subsystem) in the CFT. At large values of $E$, this entropy goes like

$$S_h(l) \approx l \sqrt{\frac{c}{3} E}. \tag{74}$$

Meaning that for a fixed value of $l$, the EE of the scarred states is much smaller (as log vs. square root) than that of a primary state of the same energy. This situation is strongly reminiscent of the behaviour of the EE of scarred vs. bulk states (i.e. states obeying the ETH) in spin systems, where they obey volume and sub-volume EE law respectively. Taking the most studied setup of a spin chain of length $L$ as an example, the EE is evaluated with respect to a spatial partitioning of the chain to two segments of length $l$ and $\bar{l} = L - l$. Here the (von Neumann) EE scales as $S \propto \log l$ and $S \propto l$ for the scarred and bulk states respectively [41].

## 5  Discussion and Outlook

In this work we have presented a systematic construction of scarred states based on dynamical symmetries in two-dimensional CFTs and their holographic dual which is associated

---

[12]Recall that we are evaluating these expressions on the Euclidean time slice corresponding to $\tau = 0$. This means that these formulas have the usual interpretation of bipartite EE between the two subregions of the CFT circle delimited by the choice of two fixed points.

[13]See also Refs. [119, 120] for the EE of the descendants (recalling that the scarred state is a particular superposition of a primary and its descendants).

with a mapping to an empty AdS space. This construction works even for states whose energies are above the BTZ threshold. The presented description might provide a useful starting point for further developments.

*Beyond coherent states.* The proposed scarred states are constructed as generalized coherent states upon the underlying Virasoro algebra and the subalgebras thereof. Here we have focused on the simplest example where the dynamical symmetry $Q \propto L_k - L_{-k}$, where one can construct a $su(1,1)$ algebra with the set $\{L_{-k}, L_0, L_k\}$. It would be interesting to study the properties of more general scarred states with symmetries $Q$ defined in (11). One obvious candidate, in analogy to states known in particular from quantum optics, would be a "squeezed" state with the charge $Q \propto L_k^2 - L_{-k}^2$. A remark is that in this case, $\big[Q, Q^\dagger\big]$ *is not* proportional to the Hamiltonian but rather satisfies the relation (5) with $L_{-k}L_k$ the conserved charge, $[L_{-k}L_k, L_0] = 0$.

Another interesting direction would be to extend the present analysis beyond pure states and study the effect of the displacement operator $D(\lambda)$ defined in (12) on mixed (including thermal) states. For the thermal state, the corresponding Loschmidt echo is defined as

$$
\begin{aligned}
|\mathcal{L}_{\text{th}}(t)|^2 &:= \text{Tr}\left(D(\lambda)\rho_{\text{th}}D^\dagger(\lambda)\mathrm{e}^{-itL_0}D(\lambda)\rho_{\text{th}}D^\dagger(\lambda)\mathrm{e}^{itL_0}\right) \\
&= \frac{1}{Z(\beta)^2}\sum_{ij}\mathrm{e}^{-\beta(E_i+E_j)}|\langle E_i| D^\dagger(\lambda)\mathrm{e}^{itL_0}D(\lambda)|E_j\rangle|^2,
\end{aligned}
\tag{75}
$$

where $\rho_{\text{th}} = \frac{1}{Z(\beta)}\sum_i e^{-\beta E_i}|E_i\rangle\langle E_i|$ includes primary states and their descendants. The contribution of primary states to this overlap can be computed using (24) and is given by the sum

$$
\frac{1}{Z(\beta)^2}\sum_{h\in\text{primary}}\mathrm{e}^{-2\left(\beta+\frac{2}{k}f(t)\right)h}\,\mathrm{e}^{-\frac{c}{6}\frac{k^2-1}{k}f(t)}, \quad \text{with}
\tag{76}
$$

$$
f(t) = \log\left|\cosh^2(k|\lambda|)\big[1 - \mathrm{e}^{ikt}\tanh^2(k|\lambda|)\big]\right|.
$$

Importantly, the Boltzmann weight is modified by an exponential contribution in the Loschmidt echo that is proportional to $h$. This leads to the effective temperature $\beta_{\text{eff}} = \beta + 2f(t)/k$. It would be interesting to understand the physical meaning of this temperature and whether this shift holds true with the inclusion of descendant states.

*Higher dimensions.* Here we have focused on the case of two dimensional CFTs which are endowed with the Virasoro algebra, which in turn allows for efficient analytical manipulations. A natural question is how the present construction extends to higher dimensions. When only global symmetries are present, it is still possible to identify dynamical symmetries obeying (2). Considering for instance a superconformal algebra in 3+1 dimensions [122], we identify the generator of dilations $D$ as the Hamiltonian. By inspection, one can identify the (global) dynamical symmetries for instance with the translation, conformal boost or supersymmetry generators $P, K$ and $Q_S, \bar{Q}_S$ respectively

$$
\begin{aligned}
[D, P] &= -P & [D, Q_S] &= -\frac{1}{2}Q_S \\
[D, K] &= K & [D, \bar{Q}_S] &= -\frac{1}{2}\bar{Q}_S.
\end{aligned}
$$

Here we drop the spatial and spin indices of the generators as they are unchanged by the commutation relations. Similarly to the remark in the previous paragraph about

"squeezed" states, the powers of the superconformal dynamical symmetries, and their corresponding algebra, obey (5).

More generally, Refs. [49, 50] proposed commutant algebras as a unifying framework for the QMBS. It is worth noting that it seems possible to identify similar algebraic structures in CFTs, even beyond 2d. One interesting candidate for that is a family of light-ray operators studied in particular in [123–125], which are in turn closely related to the $\mathcal{W}$ algebras of the 2d CFTs.

*Relation to minimal models.* Another remark is that the scarred state construction (12) can be used in other than holographic contexts. In particular, it would be interesting to evaluate the scarred state properties in minimal models corresponding typically to small central charge and their deformations breaking the integrability. To give one specific example, we consider the scarring in the deformed PXP model studied recently in Ref. [126] which, at criticality, belongs to the Ising ($c = 1/2$) universality class and where the authors could identify low-energy scars with first few descendants of the Ising model. The prescription based on the Koo-Saleur formula [127] provides an elegant way to quantitatively match the lattice operators with the field theory ones [128, 129]. It would be also interesting to compare the EE of the scars with some known results for highly excited states as studied e.g. in Ref. [130]. We leave a detailed study of this interesting opening for future work.

# Acknowledgements

We would like to thank T. Anous, M. Beşken, J. de Boer, P. Caputa, A. Dymarsky, O. Gamayun, D. Ge, H. Katsura, I. Klebanov, Y. Miao, D. Neuenfeld, K. Schoutens and C. Toldo for fruitful discussions. DL is supported by the European Research Council under the European Unions Seventh Framework Programme (FP7/2007-2013), ERC Grant agreement ADG 834878. The work of VG and WV is part of the Delta ITP consortium, a program of the Netherlands Organization for Scientific Research (NWO) funded by the Dutch Ministry of Education, Culture and Science (OCW). WV is partially supported by QuiX Quantum B.V. JM is supported by the Dutch Research Council (NWO/OCW), as part of the Quantum Software Consortium programme (project number 024.003.037).

# A  Derivation of the Loschmidt amplitude

Here we provide the details of the derivation of Eq. (29). Here $\tilde{\mathcal{L}}$ denotes the non-normalized Loschmidt amplitude,

$$
\begin{aligned}
\tilde{\mathcal{L}} &= \langle h|e^{\tilde{\alpha}^* J_-}e^{\alpha_0 J_0}e^{\tilde{\alpha} J_+}|h\rangle\, e^p \\
&= \langle h|e^{A_+ J_+}e^{\log A_0\, J_0}e^{A_- J_-}|h\rangle\, e^p \\
&= A_0^\mu e^p \\
&= \frac{1}{\left(1 - |\tilde{\alpha}|^2 e^{ikt}\right)^{2\left(\frac{h}{k} + \frac{c}{24}\frac{k^2-1}{k}\right)}} e^{iht}.
\end{aligned}
\tag{77}
$$

We have used again the expression (19) for $A_0$, here $p = -ic_k t/(2k)$. The normalized Loschmidt amplitude is then simply

$$
\mathcal{L}(t) = \frac{\tilde{\mathcal{L}}(t)}{\tilde{\mathcal{L}}(t=0)}
\tag{78}
$$

yielding the expression in (29).

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
