# Peer review of "Holographic Quantum Scars"

_SciPost Physics_

## Round 2 · Referee Report · Anonymous (Referee 1) · 2023-5-24

Report

In this paper, the authors construct a family of nonthermal eigenstates in 2D CFT by considering coherent states constructed out of Virasoro generators. They consider various signatures of thermalization, such as the return amplitude, correlation functions, and entanglement entropy. These quantities are computed using a geometric interpretation of the coherent states, and display nonthermal behavior.

This paper is an interesting addition to our understanding of thermalization in 2D CFT, I recommend it for publication after a minor edit.

Requested changes

The states constructed here are linear combinations of descendant states, and descendant states are known to be nonthermal. Given this fact, it seems to me that the novel feature of the coherent states considered here is their geometric interpretation, which allows for efficient computation of various quantities. If the authors agree with this statement, I think it may be helpful to add a sentence clarifying this in the text.

  • validity: top
  • significance: good
  • originality: good
  • clarity: top
  • formatting: perfect
  • grammar: perfect

Author:  Diego Liska  on 2023-07-05  [id 3783]

(in reply to Report 1 on 2023-05-24)

Thank you for reviewing our manuscript and for highlighting a small improvement. In response, we have added remark about thermalization in 2d CFTs (just before section 2).

---

## Round 2 · Referee Report · Anonymous (Referee 2) · 2023-6-7

Report

This work is an attempt to shed light on many-body scar states in 2d CFTs exploring the Virasoro symmetry. The topic of quantum many-body scars that generalizes ideas from classical chaos and billiards to quantum regime is very interesting, timely and may elucidate thermalisation or its lack in quantum systems. Analytically tractable examples are quite rare so this CFT setting may be of more general importance. The paper is well-written and structured and reviews relevant backgrounds to follow derivations.

Technically, they start from dynamical symmetries and recent definitions of scars exploring them. Rough idea is that Hilbert space is “fragmented” if one evolves a state by a symmetry operator that commutes with the Hamiltonian. In CFT this can be done by e.g. an appropriate combination of the Virasoro algebra generators. In most of the paper, authors consider states obtained by coherent action of the operator L_k+L_{-k} (for fixed k) on the CFT vacuum that satisfies algebra isomorphic to SL(2,R). This allows them to use the power of coherent states in computations of correlation functions or the Loschmidt amplitude and reproduce several new and known results including expectations of the CFT stress tensor and energy density. Their analysis of fixed points is elegant and quite instructive. Moreover, they discuss holographic interpretation of these coherent states using Banados geometries (by an appropriate coordinate transformation from pure AdS3). In particular, they compute holographic entanglement entropy (using Ryu-Takayanagi formula) for a single interval in these geometries. They find that entanglement entropy in their scarred state scales with the square root of the energy instead of the energy as expected for a highly excited state. Finally, they propose several interesting generalisations and applications.

This work is an interesting addition to the current literature and I recommend it for publication after small improvements:

  1. Their universal, “holographic” computation could have been also discussed purely in 2d CFTs.
  2. Is their definition of scars “unique” or there could be other routes that are more sensitive to the spectrum of a CFT and other details (e.g. rational vs irrational CFTs, large or small c?). In other words, do we always need these symmetries to have/define scars?
  3. It may be interesting to consider highest weight states instead of the vacuum (e.g. in computations of the entropy). This may highlight the need for more input about the spectrum of a CFT.
  4. Is it easy to see that entanglement entropy for arbitrary interval is “sub-extensive” (energy scaling)?
  • validity: high
  • significance: good
  • originality: good
  • clarity: high
  • formatting: good
  • grammar: excellent

Author:  Diego Liska  on 2023-07-05  [id 3782]

(in reply to Report 2 on 2023-06-07)
Category:
answer to question

Thank you for the careful reading of our manuscript and for pointing out several small improvements. Below we answer the questions which were raised in your report.

Q1: Their universal, “holographic” computation could have been also discussed purely in 2d CFTs.

A1: Yes, this computation could have also been done in the CFT using the replica trick and inserting twist operators at the end of the interval for which we compute the EE. We have added footnote 11 for some details regarding this computation.

Q2: Is their definition of scars “unique” or there could be other routes that are more sensitive to the spectrum of a CFT and other details (e.g. rational vs irrational CFTs, large or small c?). In other words, do we always need these symmetries to have/define scars?

A2: The definition is not unique. One could consider, for example, extensions of the Virasoro algebra that include current algebras (see added footnote 4). A remark is that the analogue of the Loschmidt echo for the thermal state that we mention in the discussion (equation 75) contains information about the spectrum of the CFT since it is a sum over all the primaries in the theory. However, the exact computation of such a quantity is challenging.

Q3: It may be interesting to consider highest weight states instead of the vacuum (e.g. in computations of the entropy). This may highlight the need for more input about the spectrum of a CFT.

A3: Since the scarred state only has information about its corresponding Verma module, they would, unfortunately, not have more information about the spectrum of the CFT.

Q4: Is it easy to see that entanglement entropy for arbitrary interval is “sub-extensive” (energy scaling)?

A4: We do not have an easy way to see this, but we note that analogous sub-extensive scaling occurs also in the many-body (spin) systems, as we commented on at the end of Sec. 4. Given that the scarred state is a specific superposition of the scars - descendant states - we note that results exist in the literature on the entanglement entropy of the descendants. We have now included footnote 13 and (new) references [119] and [120], which deal with such computations and which can be relatively involved.

---

## Editorial Decision

resubmitted